# mDAE : modified Denoising AutoEncoder for missing data imputation

## Abstract

This paper introduces a methodology based on Denoising AutoEncoder (DAE) for missing data imputation. The proposed methodology, called mDAE hereafter, results from a modification of the loss function and a straightforward procedure for choosing the hyperparameters. An ablation study shows on several UCI Machine Learning Repository datasets, the benefit of using this modified loss function and an overcomplete structure, in terms of Root Mean Squared Error (RMSE) of reconstruction. This numerical study is completed by comparing the mDAE methodology with eight other methods (four standard and four more recent). A criterion called Mean Distance to Best (MDB) is proposed to measure how a method performs globally well on all datasets. This criterion is defined as the mean (over the datasets) of the distances between the RMSE of the considered method and the RMSE of the best method. According to this criterion, the mDAE methodology was consistently ranked among the top methods (along with SoftImput and missForest), while the four more recent methods were systematically ranked last. The Python code of the numerical study will be available on GitHub so that results can be reproduced or generalized with other datasets and methods.

## 1    Introduction

With the rapid increase in data collection, missing values are a ubiquitous challenge across various domains. Data may be missing for several reasons. For instance, it was never collected, records were lost or merging several datasets failed. It is then generally necessary to deal with this problem before performing machine learning methods on these data. Several options are available to address this issue, including removing or recreating the missing values. Removing rows or columns containing missing data results in a considerable loss of information when missing data is distributed across multiple locations in the dataset. One usually prefers missing-data imputation, which consists of filling missing entries with estimated values using the observed data. Missing data imputation is a very active research area (Van Buuren, 2018; Little & Rubin, 2019) with more than 150 implementations available according to Mayer et al. (2021). This paper focuses on state-of-the-art imputation methods categorized as standard machine learning, deep learning or optimal transport. Methods based on standard machine learning include, among others, k-nearest neighbours (Troyanskaya et al., 2001), matrix completion via iterative soft-thresholded SVD (Mazumder et al., 2010), Multivariate Imputation by Chained Equations (Van Buuren & Groothuis-Oudshoorn, 2011) or MissForest (Stekhoven & Bühlmann, 2012). Methods based on deep learning include, among others, Generative Adversarial Networks (Goodfellow et al., 2014; Yoon et al., 2018), Variational AutoEncoders (Kingma & Welling, 2013; Ivanov et al., 2018; Mattei & Frellsen, 2019; Peis et al., 2022) and methods based on Denoising AutoEncoders (see e.g. the review of Pereira et al., 2020). One can also mention the recent works of Muzellec et al. (2020); Zhao et al. (2023) based on optimal transport.

This paper proposes a modified Denoising AutoEncoder (mDAE) dedicated to imputing missing values in numerical tabular data. AutoEncoders (AE) (Bengio et al., 2009) are artificial neural networks used to learn efficient representation of unlabeled data (encodings) and a decoding function that recreates the input data from the encoded representation. Denoising AutoEncoders (DAEs) were first proposed by Vincent et al. (2008) to recover, from noisy data, the original data without noise by corrupting the inputs of a standard AE.

For example, inputs can be corrupted by masking noise where a fixed proportion of the inputs are randomly set to 0. DAEs, initially proposed for extracting robust features in the deep learning context, have also been used for missing data imputation (see e.g. Duan et al., 2014; Gondara & Wang, 2018; Ryu et al., 2020). Indeed, DAEs, designed to recover a clean output from a noisy input, are naturally suited as an imputation method by considering missing values as a particular case of noisy input. The review of Pereira et al. (2020) covers 26 papers that use AEs and their variants (DAEs and VAEs) for the imputation of tabular data. In all these articles, except Beaulieu-Jones & Moore (2017), the reconstruction of missing data with DAEs boils down to applying a DAE to pre-imputed data (e.g., by mean imputation). Pre-imputation solves the problem of loss functions that cannot handle missing values and require all features to be complete. However, in doing so, DAEs learn to reconstruct pre-imputed values, which does not seem relevant. In the same spirit as Beaulieu-Jones & Moore (2017), we propose to deal with this problem by modifying the loss function to ignore pre-imputed missing values. We show in an ablation study that using this modified loss function in the mDAE methodology results in better reconstruction of missing values than using the unmodified loss function on pre-imputed data, thus showing the contribution of the mDAE method compared with previous imputation methods based on DAEs. Moreover, Pereira et al. (2020) points out that the vast majority of these methods do not present justifications for the decisions performed for the choice of the structure of the DAE and the choice of the hyper-parameters. Most choices are based on empirical guesses, while just a few exceptions use grid-search approaches. Following the recommendations of Pereira et al. (2020), we propose a general and reproducible grid-search methodology for choosing the hyper-parameter and the structure. Moreover, the ablation study provides recommendations for structure and hyper-parameter selection that can be used when the grid-search approach is too computationally expensive.

As mentioned above, the proposed mDAE methodology is evaluated via an ablation study to check the relevance of some of its components (modification of the loss function, choice of the hyperparameter by cross-validation, and overcomplete structure). The importance of each component is evaluated using the Root Mean Squared Error (RMSE) of reconstruction of missing values artificially added to 7 datasets from the UCI Machine Learning Repository Dua & Graff (2017). As far as we know, there are no standard benchmark datasets for missing data imputation; the 26 papers studied in the review paper of Pereira et al. (2020) almost all use different datasets. Here, we have chosen 7 of the 23 UCI Machine Learning Repository datasets recently used by Muzellec et al. (2020) for imputation methods comparison. These 7 datasets had to be all numerical (as the mDAE method is suited for numerical missing values only), with different sizes, and not too numerous (to avoid the experimental setup being time-consuming and impractical). This ablation study has two objectives. Firstly, the benefits of using the modified loss function will be shown, and thus, the mDAE method will be compared with standard DAE imputation approaches. Secondly, to show the benefit of choosing an overcomplete structure for the DAE and to verify the benefit of choosing the hyper-parameter by optimization in a grid.

After this ablation study, the mDAE method is compared with eight other imputation methods (4 based on standard machine learning and 4 based on deep learning and optimal transport) along with the reference method of mean imputation. The 10 imputation methods are compared using again the Root Mean Squared Error (RMSE) of reconstruction of missing values artificially added to the 7 datasets used in the ablation study. Moreover, as in Muzellec et al. (2020) and Zhao et al. (2023), three missing data mechanisms are considered (Missing Completely at Random, Missing At Random, Missing Not At Random). The RMSE scores of the 10 methods are then computed for each of the 7 datasets (and different percentages and mechanisms of artificial missing data). To make results easier to interpret, we propose a new criterion called Mean Distance to the Best (MDB) to measure how a method performs globally well on all datasets (for a given percentage and a given mechanism of artificial missing data). This criterion is defined as the mean (over the datasets) of the distances between the RMSE of the considered method and the RMSE of the best method. It is equal to 0 if the RMSE of the method is the best for all datasets, and it increases if the RMSE of the method is far from the RMSE of the best method, on average, over the datasets. If the proposed mDAE method sometimes gives the best RMSE score (for a given dataset), it is not true for all datasets. Considering all datasets, the MDB criterion ranks (for all configurations considered in this numerical study) three methods based on standard machine learning and the mDAE methodology in the top 4 positions (for all configurations considered in this numerical study). More precisely, the mDAE method is generally placed second or third for this criterion (alternating with the missForest method), with the SoftImput method

always ranked first. It should be noted that the 4 recent methods based on deep learning and optimal transport are always ranked in the last positions, quite far from the best methods.

According to Pereira et al. (2020), most papers that use DAEs to impute missing data report better results than SOTA (State Of The Art) methods. However, very different methodologies with different datasets and different SOTA methods were used in these papers, making a general conclusion difficult. One of the contributions of this article is to propose a comparison methodology that is as accurate and reproducible as possible so that other researchers can use it with other datasets or imputation methods using Python codes that will be available on GitHub.

## 2 The mDAE method

AutoEncoders (AE) (Bengio et al., 2009) are well-known artificial neural networks used to learn efficient representation of unlabeled data via an encoding function and to recreate the input data via a decoding function. Here, we are dealing with the special case of tabular numerical data, and we suppose that these data have been normalized so that the $p$ features have zero mean and unit variance. This normalization via feature standardization is more appropriate here than normalizing the values between 0 and 1, as is often done when using autoencoders. The input of the AE is a then set of $n$ observations $\{\mathbf{x}_1, ..., \mathbf{x}_n\}$ in $\mathbb{R}^p$ which forms the rows of a standardized data matrix $\mathbf{X} = (x_{ij})$ of dimension $n \times p$, where $p$ is the number of features. The encoding function $f_\theta$ of a basic autoencoder (see Figure 1) transforms an input $\mathbf{x}_i \in \mathbb{R}^p$ into a latent vector $\mathbf{y}_i \in \mathbb{R}^q$:

$$\mathbf{y}_i = f_\theta(\mathbf{x}_i) = s(\mathbf{W}\mathbf{x}_i + \mathbf{b}),$$

where $\mathbf{W} \in \mathbb{R}^{q \times p}$ is a weight matrix, $\mathbf{b} \in \mathbb{R}^p$ is a bias vector and $s$ is an activation function (e.g., ReLU or sigmoid). The decoding function $g_{\theta'}$ then transforms the latent vector $\mathbf{y}_i \in \mathbb{R}^q$ into an output $\mathbf{z}_i \in \mathbb{R}^p$:

$$\mathbf{z}_i = g_{\theta'}(\mathbf{y}_i) = s(\mathbf{W}'\mathbf{y}_i + \mathbf{b}'),$$

where $\mathbf{W}' \in \mathbb{R}^{p \times q}$ and $\mathbf{b}' \in \mathbb{R}^q$. Here, the activation function $s$ in the output layer must be the identity function, since we are trying to reconstruct inputs that take their values in $\mathbb{R}$. In fact, the sigmoid (resp. ReLu) activation function gives output values between 0 and 1 (resp. positive values) which is not appropriate here.

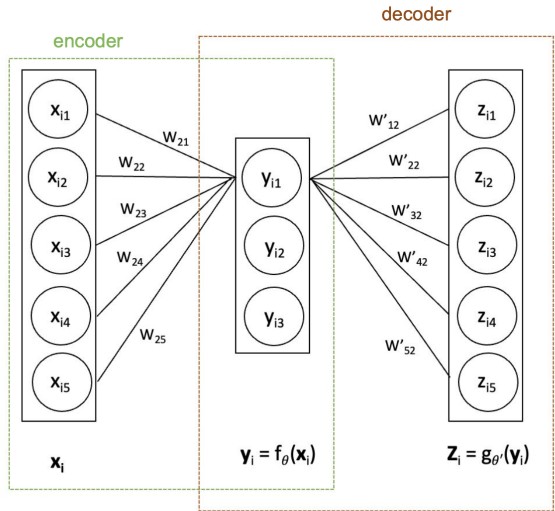

Figure 1: Scheme of a basic AutoEncoder (AE).

In general, autoencoders have more than one hidden layer and the parameters $\theta = (\mathbf{W}_1, ..., \mathbf{W}_K, \mathbf{b}_1, ..., \mathbf{b}_K)$ of the encoder and $\theta' = (\mathbf{W}'_1, ..., \mathbf{W}'_K, \mathbf{b}'_1, ..., \mathbf{b}'_K)$ of the decoder, are learned by minimization of the so called

reconstruction loss. With standardized numerical data, the reconstruction loss usually used to learn weights and biases of an autoencoder is the L2 loss defined by:

$$\mathcal{L}_{AE} = \sum_{i=1}^{n} \underbrace{\|\mathbf{x}_i - (g_{\theta'} \circ f_\theta)(\mathbf{x}_i)\|^2}_{L(\mathbf{x}_i, \mathbf{z}_i)} = \|\mathbf{X} - \mathbf{Z}\|_F^2, \tag{1}$$

where $L$ is the loss function defined here as the squared Euclidean distance between the input $\mathbf{x}_i$ and its reconstruction $\mathbf{z}_i = (g_{\theta'} \circ f_\theta)(\mathbf{x}_i)$, and $\|\mathbf{X} - \mathbf{Z}\|_F$ is the Frobenius norm between the data matrix $\mathbf{X}$ and its reconstructed matrix $\mathbf{Z}$. Note that this criterion favors the reconstruction of features (columns of $\mathbf{X}$) with high variance. It is therefore important that the data matrix $\mathbf{X}$ is standardized.

Denoising AutoEncoders (DAE) (Vincent et al., 2008) are autoencoders defined to remove noise from a given input. To do this, an autoencoder is trained to output the original data using corrupted data in the input. The masking noise, for instance, is a corrupting process where each observation $\mathbf{x}_i$ is corrupted by randomly setting a proportion $\mu$ of its components to zero. Let $N(\mathbf{x}_i)$ denotes this corrupted version of $\mathbf{x}_i$. The loss $L(\mathbf{x}_i, \mathbf{z}_i)$ is here slightly different from the one in (1) as it compares the input $\mathbf{x}_i$ with the output $\mathbf{z}_i = (g_{\theta'} \circ f_\theta)(N(\mathbf{x}_i))$ obtained with corrupted observations $N(\mathbf{x}_i)$ (see Figure 2). The $L_2$ reconstruction loss (1) writes then for DAEs:

$$\mathcal{L}_{DAE} = \sum_{i=1}^{n} \underbrace{\|\mathbf{x}_i - (g_{\theta'} \circ f_\theta)(N(\mathbf{x}_i))\|^2}_{L(\mathbf{x}_i, \mathbf{z}_i)} = \|\mathbf{X} - \mathbf{Z}\|_F^2 \tag{2}$$

Note that the proportion $\mu$ of the masking noise is a hyper-parameter that may need to be calibrated.

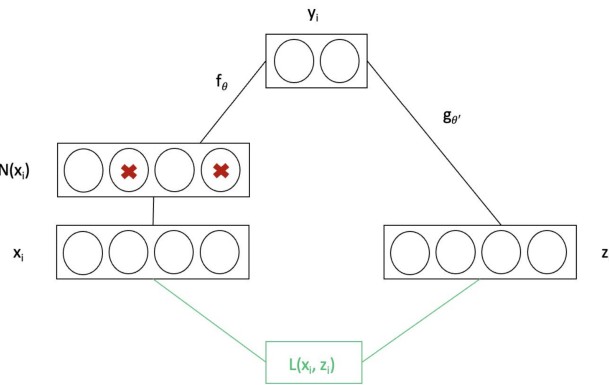

Figure 2: Scheme of a denoising AutoEncoder (DAE). Red crosses represent the values in $N(\mathbf{x}_i)$ randomly set to 0.

## 2.1 Imputing missing values using the mDAE methodology

Although DAE methods were first proposed for extracting robust features in the deep learning context Vincent et al. (2008), they have also been used for missing data imputation (see e.g. the review of Pereira et al., 2020). Indeed, as DAEs had been defined to reconstruct noisy data, they were naturally suited to reconstruct missing data, the missing data then being considered as noise.

As Pereira et al. (2020) point out in their review article, almost all works using DAEs to impute missing data boils down to applying a DAE to pre-imputed data (e.g. by mean imputation). Let $\mathbf{X}$ be now the incomplete standardized data matrix (the data matrix with missing values). Pre-imputation of $\mathbf{X}$ by the mean of each feature simply consists of replacing missing values with 0 since the data are standardized and, therefore, the feature means are all equal to 0. The pre-imputed data matrix $\tilde{\mathbf{X}}$ writes then as the projection of $\mathbf{X}$ onto the observed entries:

$$\tilde{\mathbf{X}} = P_\Omega(\mathbf{X}) = \begin{cases} x_{ij} & \text{if } (i,j) \in \Omega, \\ 0 & \text{if } (i,j) \notin \Omega. \end{cases}$$

where $\Omega$ is the set of indices $(i, j) \in \{1, ..., n\} \times \{1, ..., p\}$ where the values $\mathbf{x}_{ij}$ are not missing. The DAE is then trained to reconstruct the pre-imputed data matrix $\tilde{\mathbf{X}}$ by minimization of the reconstruction loss (2) which in this case is:

$$\mathcal{L}_{DAE} = \sum_{i=1}^{n} \|\tilde{\mathbf{x}}_i - (g_{\theta'} \circ f_\theta)(N(\mathbf{x}_i))\|^2 = \|P_\Omega(\mathbf{X}) - \mathbf{Z}\|_F^2, \tag{3}$$

where $\mathbf{Z}$ is now the reconstruction of the pre-imputed matrix $\tilde{\mathbf{X}}$. After training, the missing values in X are replaced by those reconstructed in $\mathbf{Z}$ and the imputed data matrix is:

$$\hat{\mathbf{X}} = P_\Omega(\mathbf{X}) + P_{\Omega^\perp}(\mathbf{Z}), \tag{4}$$

where $\Omega^\perp$ is the set of indices $(i, j) \in \{1, ..., n\} \times \{1, ..., p\}$ where $\mathbf{x}_{ij}$ is missing.

If using a pre-imputed matrix $\tilde{\mathbf{X}}$ solves the problem of the loss function that is unable to handle missing values, minimizing the reconstruction loss (3) learns the DAE to reconstruct zeros at the locations of the missing values, which is irrelevant (see Figure 3). Our proposal is then not only to apply a DAE to the pre-imputed data matrix as in previous works, but also to modify the reconstruction error (3) to skip these locations (see Figure 4). This methodology, herafter called mDAE, performs a DAE on standardized and pre-imputed data, using the following loss function:

$$\mathcal{L}_{mDAE} = \|P_\Omega(\mathbf{X}) - P_\Omega(\mathbf{Z})\|_F^2, \tag{5}$$

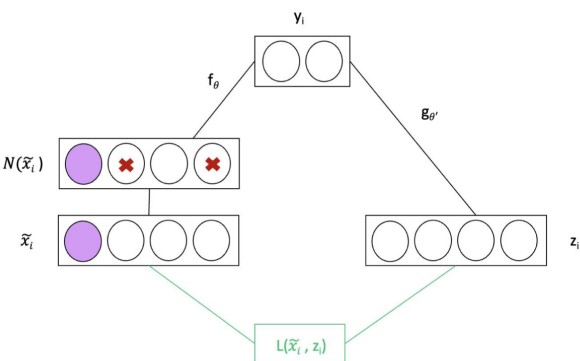

Figure 3: Scheme of a DAE directly applied on pre-imputed data. Violet dots in $\tilde{\mathbf{x}}_i$ represent the missing values set to 0. Red crosses in $N(\tilde{\mathbf{x}}_i)$ represent the values randomly set to 0.

## 2.2 Choice of the hyper-parameter $\mu$

The hyper-parameter $\mu$ of the mDAE methodology for missing values imputation, is the proportion of zeros used to corrupt the data with the masking noise (red crosses in $N(\tilde{\mathbf{x}}_i)$ in Figure 4). This hyper-parameter can be chosen randomly in a grid of values $\mu$ in $[0, 1]$. Alternatively, it can be chosen through an optimized procedure to minimize an error of reconstruction of the missing values. For that purpose, the non-missing values of the original data are split into two sets: a training set to learn the parameters and a validation set to estimate the error of reconstruction of missing values. Let $V \subset \Omega$ be the subset of indices $(i, j)$ of the validation set, drawn randomly from the set of observed entries $\Omega$. For each value of $\mu$ in the grid, the error of reconstruction of the missing values is estimated using the following procedure:

1. The parameters of the mDAE are learned on the training set $\Omega \setminus V$ by minimization of the reconstruction loss:

$$\mathcal{L}_{mDAE} = \|P_{\Omega \setminus V}(\mathbf{X}) - P_{\Omega \setminus V}(\mathbf{Z})\|_F^2, \tag{6}$$

where $\Omega \setminus V$ is the set of observed entries minus those drawn at random for the validation.

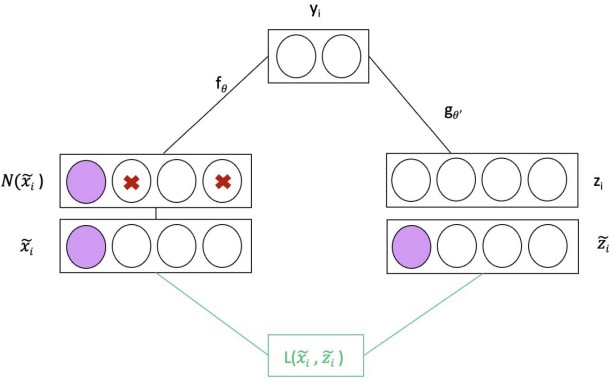

Figure 4: Scheme of a mDAE. Violet dots in $\tilde{\mathbf{x}}_i$ represent the missing values set to 0. Violet dots in $\tilde{\mathbf{z}}_i$ represent the predicted values set to 0. Red crosses in $N(\tilde{\mathbf{x}}_i)$ represent the values randomly set to 0

2. The mean squared error (MSE) of reconstruction of the missing values is estimated on the validation set by:

$$MSE_{val} = \frac{1}{|V|}\|P_V(\mathbf{X}) - P_V(\mathbf{Z})\|_F^2. \tag{7}$$

where $\mathbf{Z}$ is the matrix reconstructed with the mDAE learned on the training set $\Omega \setminus V$ and $|V|$ is the cardinal of the validation set.

The previous two steps are repeated $B$ times (for the $B$ draws of missing values) and the mean of the errors of reconstruction of the missing values is performed to get a more robust estimation.

## 2.3 Choice of the structure

Two families of structures are known for autoencoders. The undercomplete case where the hidden layer is smaller than the input layer and the overcomplete case where it is bigger. If overcomplete structure is not relevant with autoencoders, it is well-known that denoising autoencoders work well with overcomplete structures. Here, a grid of 6 simple structures (2 undercomplete and four overcomplete) is suggested to choose the "best" structure when using the mDAE method (see Figure 5). For each structure in this grid, the error of reconstruction of the missing values is estimated on validation data, using the same procedure as for the selection of the hyper-parameter $\mu$ (see section 2.2). Ideally, the hyper-parameter $\mu$ and the structure should be chosen simultaneously by exhaustively considering all possible combinations. However, alternative grid search exists, for instance, by sampling a given number of candidates from the parameter space.

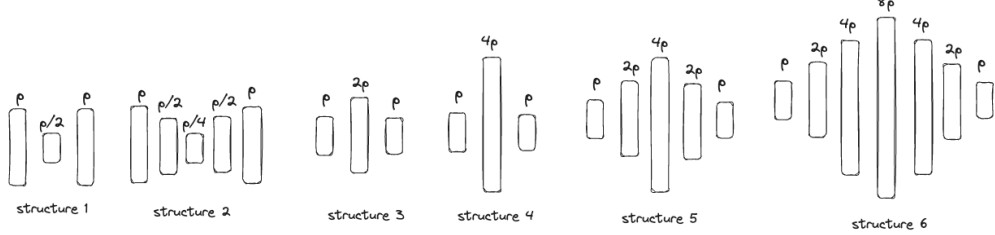

Figure 5: A grid of 6 simple structures where $p$ is the number of units of the input layer.

# 3 Numerical study

The first part of this numerical study concerns the properties of the mDAE methodology. More specifically, an ablation study is conducted to verify the relevance of the choices made to construct this methodology. The second part compares the mDAE method with other well-known or more recent methods for imputing missing data.

All comparisons are made using seven complete tabular datasets (without missing values) chosen among 23 datasets of the UCI Machine Learning Repository, recently used by Muzellec et al. (2020) for imputation methods comparison. These 7 datasets (see Table 1) have been chosen to be all numerical (as the mDAE method is suited for numerical missing values only), with different sizes and not too numerous (to avoid the experimental setup being time-consuming and impractical).

Table 1: The seven datasets used in the numerical study

| Names | Abreviations | Rows | Columns |
|---|---|---|---|
| Breast cancer diagnostic | breast | 569 | 30 |
| Connectionist bench sonar | sonar | 208 | 60 |
| Ionosphere | iono | 351 | 34 |
| Blood transfusion | blood | 748 | 4 |
| Seeds | seeds | 210 | 7 |
| Climate model crashes | climate | 540 | 18 |
| Wine quality red | wine | 1599 | 10 |

To evaluate the imputation methods, a certain proportion of each dataset is first artificially replaced by missing values. The artificial missing values are drawn using either the MAR (Missing At Random), the MCAR (Missing Completely At Random) or the MNAR (Missing Not At Random) mechanism (see e.g. Rubin, 1976). Note that the MCAR and MAR missing values were generated using a logistic masking model as implemented in the GitHub repository of Muzellec.

Then, for a given mask $\Omega^\perp$ of artificial missing values, the performance of the method is evaluated using the Root Mean Squared Error (RMSE) between the initial data matrix $\mathbf{X}$ and the reconstructed data matrix $\mathbf{Z}$ on $\Omega^\perp$:

$$RMSE = \sqrt{\frac{1}{|\Omega^\perp|} \|P_{\Omega^\perp}(\mathbf{X}) - P_{\Omega^\perp}(\mathbf{Z})\|_F^2}, \tag{8}$$

where $|\Omega^\perp|$ is the number of artificial missing values. To get more robust results, the process is repeated $B$ times with $B$ sets of artificial missing values drawn randomly using one of the three generation mechanisms. Finally, a method is evaluated by the mean and standard deviation of the $B$ values of RMSE obtained with a certain proportion of artificial missing data and a certain mechanism of missing values (MAR, MCAR or MNAR).

Note that all the results presented in this section are reproducible using Python code, which will be available on GitHub.

## 3.1 Ablation study of the mDAE methodology

An ablation study is a methodology used to evaluate the importance of different components of an algorithm, by comparing the results obtained with and without this component. Here, the following components of the mDAE methodology are studied:

- the use of the modified reconstruction loss (5) rather than the standard loss (3),

- the use of an optimized value of the hyper-parameter $\mu$ (as described section 2.2) rather than a value chosen randomly in $[0, 1]$,

- the use of an overcomplete structure (the 5th structure in Figure 5) rather than an undercomplete structure (the 2nd structure in Figure 5).

Table 2 shows the results of the ablation study for the seven datasets and 20% of MCAR artificial missing values. The mean value over the $B$ sets of artificial missing values ($\pm$ the standard deviation) of the RMSE of reconstruction of the artificial missing values is calculated for each dataset with the mDAE method, with the method deprived of its modified loss function (i.e. with a standard $L_2$ loss function), with the method deprived of its optimized choice of $\mu$ (i.e. with a random choice), with the method deprived of its overcomplete structure (i.e. with an under complete structure). Each time, the loss of imputation quality (i.e. the increase of the mean RMSE) is measured between the mDAE without one of the three components (the modified loss, an optimized choice of $\mu$ or an overcomplete structure) and the complete mDAE. For instance, for the breast cancer dataset, using the standard $L_2$ loss increases the mean RMSE of $46.99\% = \frac{0.685 - 0.466}{0.466}$.

| Method | breast | climate | sonar | iono | seeds | wine | blood |
|---|---|---|---|---|---|---|---|
| **mDAE** | **0.466 ± 0.016** | 1.007 ± 0.007 | **0.656 ± 0.007** | **0.776 ± 0.018** | **0.496 ± 0.022** | **0.790 ± 0.030** | **0.701 ± 0.059** |
| mDAE w/o modified loss | 0.685 ± 0.036 (46.996%) | **1.005 ± 0.008** (-0.199%) | 0.988 ± 0.013 (50.610%) | 0.808 ± 0.020 (4.124%) | 0.587 ± 0.028 (18.347%) | 0.828 ± 0.034 (4.810%) | 0.755 ± 0.058 (7.703%) |
| mDAE w/o optimal $\mu$ | 0.501 ± 0.043 (7.511%) | 1.030 ± 0.013 (2.284%) | 0.682 ± 0.049 (3.963%) | 0.802 ± 0.039 (3.351%) | 0.514 ± 0.054 (3.629%) | 0.853 ± 0.033 (7.975%) | 0.710 ± 0.055 (1.284%) |
| mDAE w/o overcomplete | 0.500 ± 0.011 (7.296%) | 1.147 ± 0.013 (13.903%) | 0.699 ± 0.008 (6.555%) | 0.808 ± 0.025 (4.124%) | 0.671 ± 0.209 (35.282%) | 0.932 ± 0.045 (17.975%) | 0.960 ± 0.140 (36.947%) |

Table 2: Mean RMSE of reconstruction ($\pm$ the standard deviation) for $B = 8$ random draws of 20% of MCAR artificial missing values. First row : results of the mDAE method (with the modified loss, the optimal choice of the hyper-parameter $\mu$ and with an overcomplete structure). Second row : results without (w/o) the modified loss (with the standard $L_2$ loss instead). Third row : results without (w/o) the optimal choice of $\mu$ (with random choice of $\mu$ instead). Fourth row : results without (w/o) overcomplete structure (with an undercomplete structure instead). The results in brackets are the growth rate of the average RMSE when the component under consideration is removed.

The first row in Table 2 shows that the mDAE methodology with its three components (modified loss function, optimized choice of $\mu$ and overcomplete structure 5 of Figure 5) constantly reconstructs missing data better, except for climate data, where modifying the loss function does not improve the results. It should be noted that this last result is consistent with those obtained by Muzellec et al. (2020), who found that, for the climate dataset (and 30% MCAR), the 5 imputation methods compared in their article gave no better results (in terms of RMSE) than imputation by the mean.

The second row in Table 2 shows the improvement (in terms of RMSE) when using the modified loss function rather than simply a DAE on pre-imputed data (as in previous works). Not using the modified loss function increases the RMSE for the breast and seeds datasets by up to 50%, thus showing the contribution of the mDAE methodology.

The third row shows that using a random value of the hyper-parameter $\mu$ rather than an optimized one deteriorates the imputation quality for all datasets, but to a lesser extent (between 1 and 8% increase in RMSE). The gain obtained by choosing the best $\mu$ in a grid rather than randomly in $[0, 1]$ is insignificant. This is an important result, as it allows the user to randomly choose the hyper-parameter $\mu$ in $[0, 1]$ to save computation time when necessary.

The fourth row shows that using an undercomplete structure rather than an overcomplete one clearly increases the RMSE to around 35% for two of the seven datasets. The choice of the structure is a central issue when using DAEs. This result can, therefore, be used to recommend the choice of an overcomplete structure and avoid the search for an optimal structure in a grid. Figure 6 completes the results of Table 2 by

looking at the results for the 6 different structures given Figure 5. It shows that here, the two undercomplete structures always give poorer results than the four overcomplete ones.

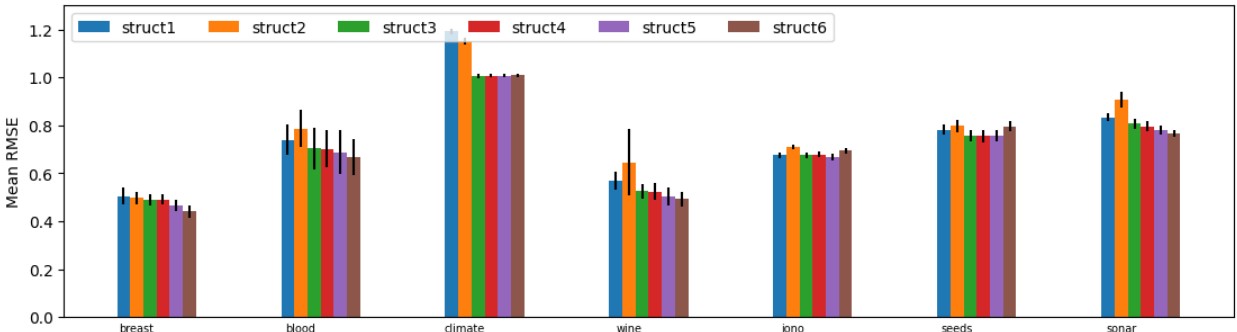

Figure 6: Mean RMSE of reconstruction ($\pm$ the standard deviation) for $B = 12$ random draws of 20% of MCAR artificial missing values and 6 different structures of mDAE. The two first structures are undercomplete, the three last are overcomplete (see Figure 5).

Finally, the results of ablation studies for other types of artificial missing values (MAR and MNAR) and other proportions of artificial missing values (20% and 40%) are given in Appendix A. These results confirm the importance of the modification of the loss function, the importance of choosing an overcomplete structure, and the more relative importance of choosing $\mu$ in a grid search rather than a random one.

## 3.2 Comparison with other methods

This section compares the mDAE method with four relatively classic and four more recent methods (see Table 3). The four first methods are KNN ((Troyanskaya et al., 2001)) where missing values are replaced by a weighted average of the $k$-nearest neighbours, SoftImput (Mazumder et al., 2010) based on iterative soft-thresholded SVD, and two iterative chained equation methods (Van Buuren & Groothuis-Oudshoorn, 2011) which model features with missing values as a function of the others: the missForest method (Stekhoven & Bühlmann, 2012) is based on Random Forests and the BayesianRidge method is based on ridge regressions, to estimate at each step the regression functions. The four others (more recent) methods in Table 3 are GAIN (Yoon et al., 2018) which is an adaptation of Generative Adversarial Networks (GAN) (Goodfellow et al., 2014) to impute missing data, MIWAE Mattei & Frellsen (2019) which is an adaptation of Variational AutoEncoders (VAE) (Kingma & Welling, 2013), and two methods using optimal transport: the algorithm called Batch Sinkhorn Imputation proposed by Muzellec et al. (2020), and the method TDM proposed by Zhao et al. (2023).

For KNN and SoftImpute, the hyperparameters are selected through cross-validation. According to the implementations used for the two chained equation methods, the hyperparameters of the Bayesian ridge regressions are estimated during the fits of the model. The hyperparameters of the Random Forests are 100 trees, and all features are considered when looking for the best split (i.e., bagged trees). The hyperparameter settings recommended in the corresponding papers and implementations are used for the four last methods. For the mDAE method, the settings studied section 3.1 (choice of $\mu$ by crossvalidation and the overcomplete structure 5 of Figure 5) are used. More favourable settings for the mDAE method would have been to select $\mu$ and the structure by cross-validation on all possible parameter combinations. This approach was not adopted for computation time reasons in this numerical study.

---

[1] Available in the class KNNImputer, https://scikit-learn.org/stable/api/sklearn.impute.html

[2] https://github.com/BorisMuzellec/MissingDataOT

[3] Available in the class IterativeImputer, https://scikit-learn.org/stable/api/sklearn.impute.html

[4] https://github.com/jsyoon0823/GAIN

[5] https://github.com/pamattei/miwae

[6] https://github.com/hezgit/TDM

Table 3: The methods used in the numerical study

| Names | Abreviations |
|---|---|
| $k$-nearest neighbors[1] | knn |
| SoftImput[2] | si |
| missForest[3] | rf |
| BayesianRidge[3] | br |
| Generative Adversarial Imputation Network[4] | gain |
| Missing Data Importance Weighted Autoencoders[5] | miwae |
| Batch Sinkhorn Imputation[2] | skh |
| Transformed Distribution Matching for missing value imputation[6] | tdm |

With these settings of hyperparameters, the eight methods of Table 3, as well as the mDAE method and the basic mean imputation method, are compared in Figure 7 on the 7 datasets and 20% of MCAR artificial missing values. The mean value (± the standard deviation) of the RMSE of reconstruction of the artificial missing values is plotted for each dataset and each method.

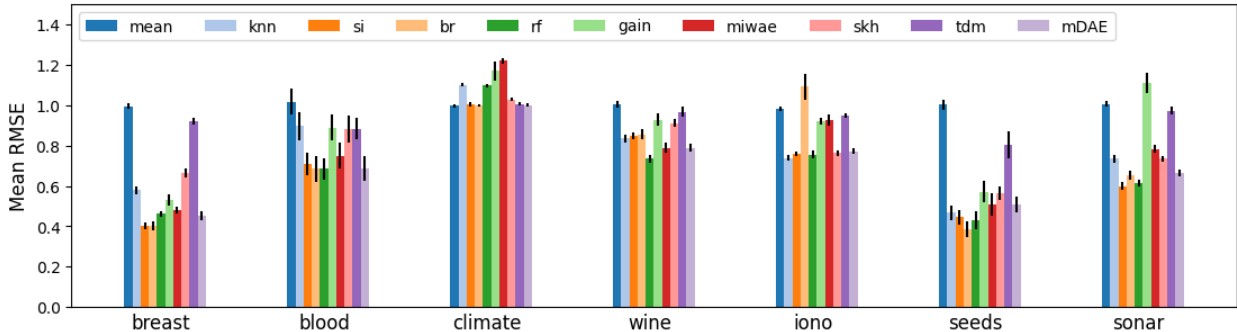

Figure 7: Mean RMSE of reconstruction (± the standard deviation) for $B = 12$ random draws of 20% of MCAR artificial missing values.

We note in Figure 7 that certain methods like SoftImpute (si), missForest (rf) or mDAE work reasonably well on all datasets (no dataset where the RMSE value is much worse than others). It can also be noted that the mDAE method gives better or equivalent results on the 7 datasets than the four methods based on neural networks and optimal transport (gain, miwae, skh and tdm).

But no method always wins. In order to measure how a method performs globally well on several datasets, we propose to use a new metric called Mean Distance to the Best (MDB) hereafter. If $I$ denotes the number of datasets and $J$ the number of methods, the MDB of a method $j$ is defined by:

$$MDB(j) = \frac{1}{I} \sum_{i=1}^{I} \left( R_{ij} - \min_{\ell=1...J} R_{i\ell} \right) \tag{9}$$

where $R_{ij}$ is the RMSE obtained with the method $j$ on the dataset $i$. $MDB(j)$ interprets as the mean (over the datasets) of the distances between the RMSE of the method $j$ and the RMSE of the best method. It is equal to 0 if the method $j$ is the best for all datasets. It increases if the quality of the method $j$ is far from the quality of the best method, on average over the datasets.

Figure 8 shows the MDB obtained with 20% of artificial MCAR missing values and the quality (the RMSE) of the methods plotted Figure 7 . This figure shows that the two best methods according to this criterion are SoftImput (si) and missForest (rf). The mDAE method is the 3rd best method. The Figures 9, 10 and 11 in Appendix B show the results with 40% of artificial MCAR missing values, and with 20% or 40% of

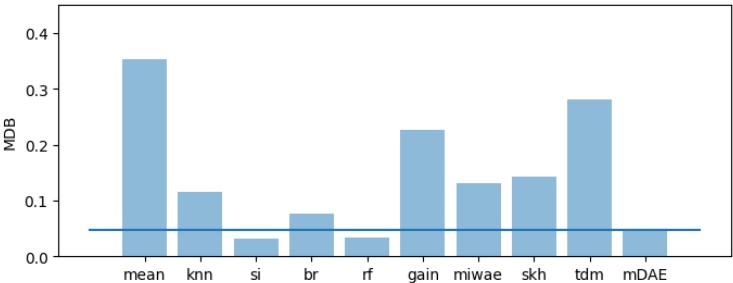

Figure 8: Mean Distance to the Best (MDB) obtained with 20% of MCAR artificial missing values.

MAR and MNAR missing values. With these different proportions and types of missing data, the top four remains SoftImput, missForest, mDAE and BayesianRidge. SofImpute is always in first place, tied once (40% MAR) with mDAE. The mDAE and missForest methods are generally one or the other in second and third position. The same type of results can be obtained for other proportions of artificial missing values using Python codes that will be available on GitHub. The results obtained, for instance, with 10% of MCAR artificial missing values (see Figure 12 in Appendix B) confirm that the mDAE methodology and the three methods SoftImput, BayesianRidge, missForest, rank (according to the MDB criterion) ahead of the KNN, ahead of the two methods gain and miwae (based on deep learning) and ahead the two methods skh and tdm (based on optimal transport). The poor results of the four more recent methods based on neural networks and optimal transport (gain, miwae, skh and tdm) can be explained by the difficulty of choosing the best hyper-parameters, the default configurations recommended by the authors having been used here.

## 4 Conclusion

This article proposes a methodology for missing data imputation, based on DAE, as well as a procedure for choosing the hyper-parameters (the proportion of noise $\mu$ and the structure of the network). An ablation study of this method was performed with different datasets, different types and proportions of missing data. It showed the relatively small improvement of the results when the hyper-parameter $\mu$ is chosen by cross-validation rather than randomly. On the contrary, using an overcomplete rather than an undercomplete network seems appropriate. A specific study is still required to confirm this result, which would enable to recommend the use of a random $\mu$ and an overcomplete structure.

Then, a numerical study compared the proposed mDAE method with eight other standard or recent missing values imputation methods. The results showed the good behavior of SofImput, mDAE and missForest. A new criterion called Mean Distance to the Best (MDB) was used to compare the methods globally over all the considered datasets and to rank them. The four most recent methods based on deep learning and optimal transport were systematically found in the last four positions for all types and proportions of artificial missing values. One might think these methods give better results with image or natural language processing data. This should be tested more thoroughly. The Python code for this numerical comparison will be made available on GitHub so that it can be reproduced with other datasets or completed with other methods.

Finally, the specific features of the mDAE method should make it possible to consider block-wise missing values by imposing a block-wise structuring of the masking noise. This type of missing data is frequent, for instance, with Electronic health records, longitudinal studies or time series data, where failures in sensors and communication can result in a loss of multiple consecutive data points.

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

# A    Appendix

| Method | breast | climate | sonar | iono | seeds | wine | blood |
|---|---|---|---|---|---|---|---|
| **mDAE** | **0.535 ± 0.012** | **1.005 ± 0.009** | **0.735 ± 0.011** | **0.793 ± 0.012** | **0.537 ± 0.026** | **0.862 ± 0.029** | 0.761 ± 0.053 |
| mDAE w/o modified loss | 0.829 ± 0.021 (54.953%) | 1.006 ± 0.008 (0.100%) | 1.007 ± 0.007 (37.007%) | 0.880 ± 0.013 (10.971%) | 0.741 ± 0.029 (37.989%) | 0.927 ± 0.027 (7.541%) | 0.844 ± 0.052 (10.907%) |
| mDAE w/o optimal $\mu$ | 0.538 ± 0.028 (0.561%) | 1.023 ± 0.018 (1.791%) | 0.764 ± 0.041 (3.946%) | 0.832 ± 0.035 (4.918%) | 0.563 ± 0.052 (4.842%) | 0.885 ± 0.055 (2.668%) | **0.746 ± 0.054** (-1.971%) |
| mDAE w/o overcomplete | 0.548 ± 0.014 (2.430%) | 1.159 ± 0.014 (15.323%) | 0.774 ± 0.013 (5.306%) | 0.845 ± 0.016 (6.557%) | 0.756 ± 0.203 (40.782%) | 0.959 ± 0.028 (11.253%) | 0.849 ± 0.106 (11.564%) |

Table 4: Mean RMSE of reconstruction (± the standard deviation) for $B = 8$ random draws of 40% of MCAR artificial missing values.

| Method | breast | climate | sonar | iono | seeds | wine | blood |
|---|---|---|---|---|---|---|---|
| **mDAE** | 0.484 ± 0.039 | 1.009 ± 0.012 | **0.657 ± 0.033** | **0.834 ± 0.029** | **0.468 ± 0.057** | **0.829 ± 0.042** | **0.613 ± 0.187** |
| mDAE w/o modified loss | 0.812 ± 0.066 (67.769%) | **1.005 ± 0.011** (-0.396%) | 0.978 ± 0.026 (48.858%) | 0.898 ± 0.028 (7.674%) | 0.682 ± 0.088 (45.726%) | 0.892 ± 0.061 (7.600%) | 0.839 ± 0.299 (36.868%) |
| mDAE w/o optimal $\mu$ | **0.482 ± 0.038** (-0.413%) | 1.033 ± 0.015 (2.379%) | 0.686 ± 0.042 (4.414%) | 0.880 ± 0.055 (5.516%) | 0.485 ± 0.071 (3.632%) | 0.888 ± 0.090 (7.117%) | 0.637 ± 0.225 (3.915%) |
| mDAE w/o overcomplete | 0.521 ± 0.035 (7.645%) | 1.161 ± 0.013 (15.064%) | 0.716 ± 0.030 (8.980%) | 0.899 ± 0.046 (7.794%) | 0.830 ± 0.294 (77.350%) | 0.974 ± 0.065 (17.491%) | 0.967 ± 0.345 (57.749%) |

Table 5: Mean RMSE of reconstruction (± the standard deviation) for $B = 8$ random draws of 20% of MAR artificial missing values.

| Method | breast | climate | sonar | iono | seeds | wine | blood |
|---|---|---|---|---|---|---|---|
| **mDAE** | **0.510 ± 0.032** | **1.005 ± 0.006** | **0.718 ± 0.017** | **0.804 ± 0.018** | **0.511 ± 0.040** | **0.830 ± 0.021** | **0.658 ± 0.090** |
| mDAE w/o modified loss | 0.854 ± 0.045 (67.451%) | 1.005 ± 0.008 (0.000%) | 1.000 ± 0.024 (39.276%) | 0.891 ± 0.025 (10.821%) | 0.821 ± 0.052 (60.665%) | 0.916 ± 0.027 (10.361%) | 0.893 ± 0.155 (35.714%) |
| mDAE w/o optimal $\mu$ | 0.546 ± 0.059 (7.059%) | 1.033 ± 0.021 (2.786%) | 0.766 ± 0.033 (6.685%) | 0.846 ± 0.046 (5.224%) | 0.537 ± 0.052 (5.088%) | 0.925 ± 0.047 (11.446%) | 0.668 ± 0.099 (1.520%) |
| mDAE w/o overcomplete | 0.526 ± 0.023 (3.137%) | 1.149 ± 0.015 (14.328%) | 0.780 ± 0.024 (8.635%) | 0.868 ± 0.028 (7.960%) | 0.743 ± 0.230 (45.401%) | 1.018 ± 0.140 (22.651%) | 0.989 ± 0.232 (50.304%) |

Table 6: Mean RMSE of reconstruction (± the standard deviation) for $B = 8$ random draws of 40% of MAR artificial missing values.

| Method | breast | climate | sonar | iono | seeds | wine | blood |
|---|---|---|---|---|---|---|---|
| **mDAE** | **0.486 ± 0.029** | 1.001 ± 0.006 | **0.684 ± 0.013** | **0.829 ± 0.027** | **0.503 ± 0.032** | **0.805 ± 0.033** | **0.738 ± 0.176** |
| mDAE w/o modified loss | 0.795 ± 0.048 (63.580%) | **1.000 ± 0.008** (-0.100%) | 1.005 ± 0.019 (46.930%) | 0.890 ± 0.033 (7.358%) | 0.682 ± 0.084 (35.586%) | 0.864 ± 0.043 (7.329%) | 0.943 ± 0.248 (27.778%) |
| mDAE w/o optimal $\mu$ | 0.518 ± 0.050 (6.584%) | 1.024 ± 0.011 (2.298%) | 0.698 ± 0.030 (2.047%) | 0.836 ± 0.021 (0.844%) | 0.531 ± 0.025 (5.567%) | 0.839 ± 0.076 (4.224%) | 0.772 ± 0.213 (4.607%) |
| mDAE w/o overcomplete | 0.521 ± 0.025 (7.202%) | 1.156 ± 0.016 (15.485%) | 0.742 ± 0.028 (8.480%) | 0.893 ± 0.055 (7.720%) | 0.686 ± 0.229 (36.382%) | 0.965 ± 0.091 (19.876%) | 0.950 ± 0.288 (28.726%) |

Table 7: Mean RMSE of reconstruction (± the standard deviation) for $B = 8$ random draws of 20% of MNAR artificial missing values.

| Method | breast | climate | sonar | iono | seeds | wine | blood |
|---|---|---|---|---|---|---|---|
| **mDAE** | **0.543 ± 0.030** | **1.005 ± 0.006** | **0.738 ± 0.018** | **0.798 ± 0.018** | **0.524 ± 0.031** | **0.873 ± 0.042** | **0.737 ± 0.102** |
| mDAE w/o modified loss | 0.866 ± 0.043 (59.484%) | 1.007 ± 0.007 (0.199%) | 0.998 ± 0.016 (35.230%) | 0.893 ± 0.011 (11.905%) | 0.800 ± 0.039 (52.672%) | 0.950 ± 0.047 (8.820%) | 0.885 ± 0.109 (20.081%) |
| mDAE w/o optimal $\mu$ | 0.574 ± 0.048 (5.709%) | 1.016 ± 0.012 (1.095%) | 0.750 ± 0.039 (1.626%) | 0.830 ± 0.035 (4.010%) | 0.565 ± 0.051 (7.824%) | 0.922 ± 0.039 (5.613%) | 0.750 ± 0.115 (1.764%) |
| mDAE w/o overcomplete | 0.559 ± 0.024 (2.947%) | 1.158 ± 0.013 (15.224%) | 0.796 ± 0.025 (7.859%) | 0.859 ± 0.020 (7.644%) | 0.827 ± 0.236 (57.824%) | 1.000 ± 0.034 (14.548%) | 0.927 ± 0.082 (25.780%) |

Table 8: Mean RMSE of reconstruction (± the standard deviation) for $B = 8$ random draws of 40% of MNAR artificial missing values.

## B  Appendix

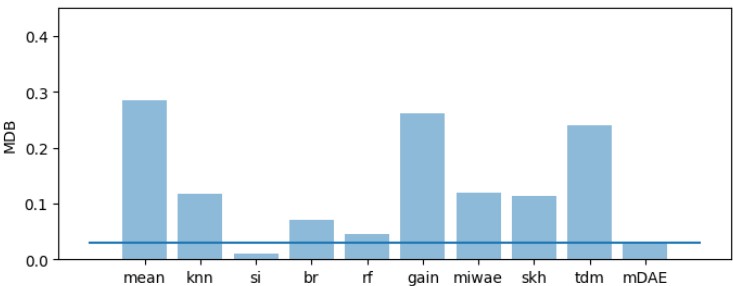

Figure 9: Mean Distance to the Best (MDB) obtained with 40% of MCAR artificial missing values.

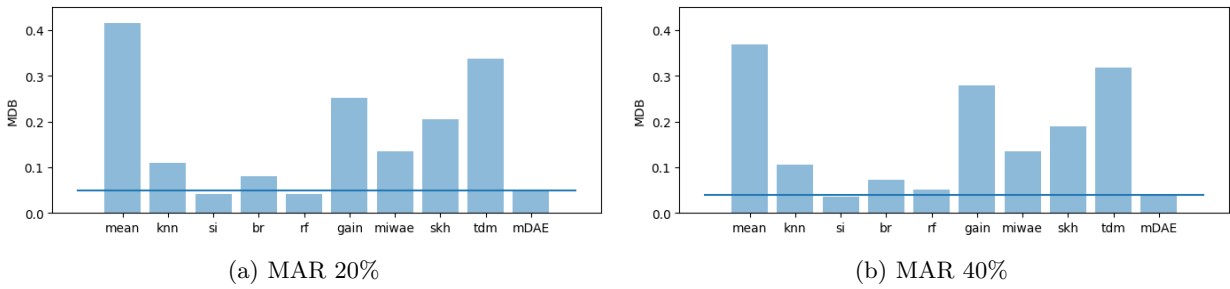

(a) MAR 20%                                          (b) MAR 40%

Figure 10: Mean Distance to the Best (MDB) obtained with MAR artificial missing values.

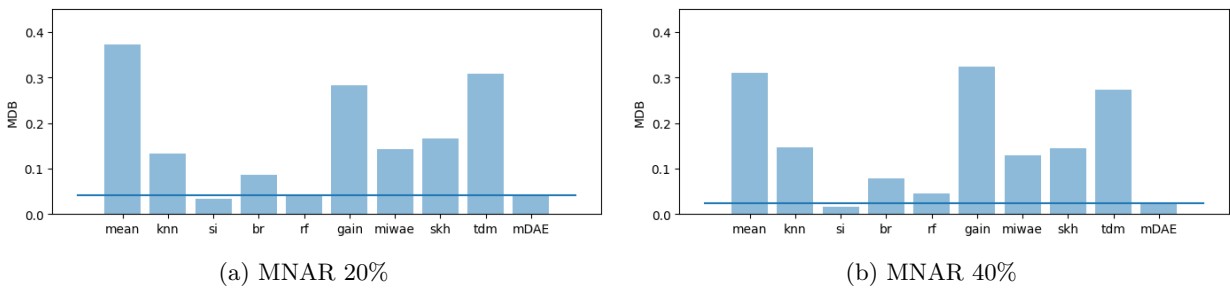

(a) MNAR 20%                                         (b) MNAR 40%

Figure 11: Mean Distance to the Best (MDB) obtained with MNAR artificial missing values.

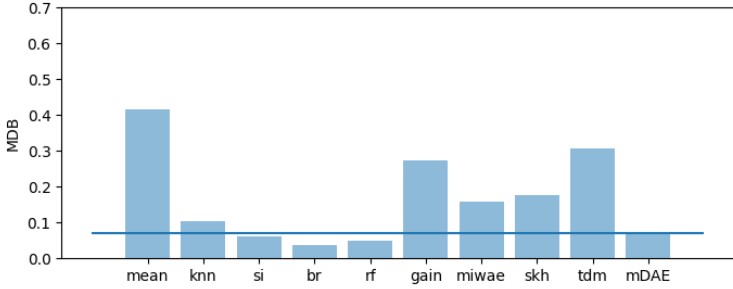

Figure 12: Mean Distance to the Best (MDB) obtained with 10% MCAR artificial missing values.

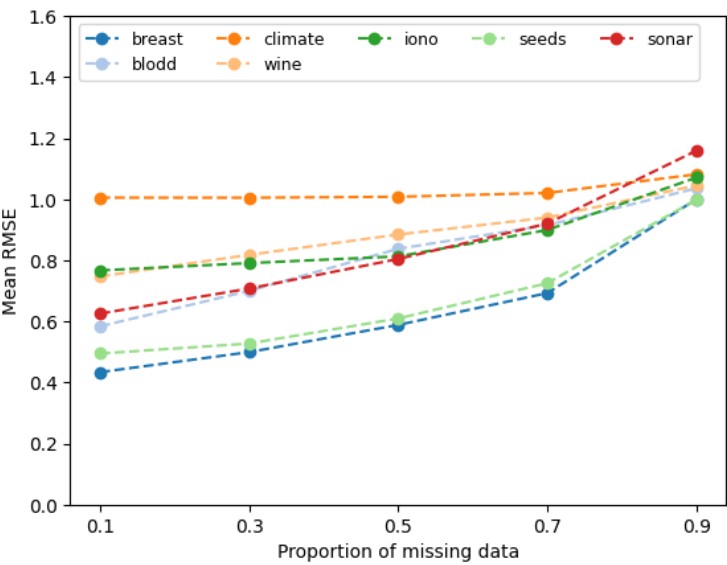

Figure 13

