# OpenReview forum: "mDAE : modified Denoising AutoEncoder for missing data imputation"
_TMLR — Rejected by TMLR_

### Review · Reviewer_i3J1 · 2024-08-05

**Summary Of Contributions:**

The authors propose using a denoising autoencoder approach to improve the performance of data imputation approaches. The contributions of this paper can be summarised as follows:

1.	Introduces a modified Denoising AutoEncoder (mDAE) method for missing data imputation
2.	Conducts ablation studies to demonstrate the relevance of the method's components
3.	Compares mDAE to 8 other imputation methods on a few UCI datasets
4.	Proposes a "Mean Distance to Best" (MDB) metric for comparing imputation methods across datasets

**Audience:**

Yes

**Claims And Evidence:**

No

**Requested Changes:**

A couple of changes to consider and a question I have that I would like the authors to reflect on are:

Suggested Changes:

1.	Please include more theoretical motivation/analysis for the method
2.	Please expand the evaluation or provide your motivation as to why these specific datasets have been selected. Are there no standard benchmark datasets for imputation?
3.	Can you please investigate the reasons for the poor performance of some recent deep learning methods; see my comment above as well regarding providing a more critical evaluation and discussion of the results.
4.	Please consider additional ablation studies on network architecture choices

Questions: \
1.	How does the mDAE method compare to other autoencoder-based imputation approaches? Authors have cited a couple of methods but they are not included in the comparisons.

**Strengths And Weaknesses:**

Strengths: \
1.	Thorough empirical evaluation of multiple datasets and missing data scenarios \
2.	Ablation study provides insights into the importance of different components, largely the modified loss function which aims to avoid learning to reconstruct zeros at the locations of missing values. \
3.	Comparison with both non-DL and DL imputation methods, such as random forests, KNN, and others. \
4.	They use a mean distance to the best metric (MDB) which allows for a more holistic and cross-dataset comparison of the method, rather than evaluating each dataset as a separate case.  The concept is simple and is sort of a mean distance between the best method and a current method that normalises things \
5.	Code will be made available for reproducibility and has been provided as supplementary material in this submission too

Weaknesses: \
1.	Limited theoretical analysis or justification for the method, which includes very little information on other DAE methods. There are a couple of papers cited in the introduction but there is very little motivation as to why DAEs, what other DAE approaches have done in this area, and also comparison against them \
2.	I see limited motivation as to why these specific datasets were selected from within UCI \
3.	Given that some recent deep learning methods perform poorly, the paper should have a more critical approach and explore/discuss the findings in a more technical way as well. \
4.	The paper is a bit shallow and includes several sections that are trivial, e.g. explaining autoencoders and denoising autoencoders. \
5.	I think that the mDAE method is more of a methodology around how to treat missing values in datasets rather than a new DAE method per se

---

> ### Author Response · Authors · 2024-08-27
> **Response to reviewer i3J1**
>
> We thank the reviewer i3J1 for his evaluation of the paper and his insightful comments. Our answers to the reviewers' questions/comments are given below. Note that changes in the revised manuscript are shown in blue.
>
> > * Weakness 1 : Limited theoretical analysis or justification for the method, which includes very little information on other DAE methods. There are a couple of papers cited in the introduction but there is very little motivation as to why DAEs, what other DAE approaches have done in this area, and also comparison against them
> > * Question of the requested changes : How does the mDAE method compare to other autoencoder-based imputation approaches? Authors have cited a couple of methods but they are not included in the comparisons.
> > * Suggested changes 1 : Please include more theoretical motivation/analysis for the method.
>
> We agree with these comments. We have modified the second paragraph of the introduction to clarify why DAEs are relevant for imputing missing data and the difference between the proposed mDAE methodology and existing imputation methods based on DAEs.
> Section 2.1 has been modified to present the theoretical motivation for the mDAE methodology more clearly.  We also explain more clearly (both in the introduction and section 3.1) that the ablation study makes it possible to compare numerically the mDAE methodology with other existing methods that apply DAEs directly to pre-imputed data.
>
> > * Weakness 2: I see limited motivation as to why these specific datasets were selected from within UCI.
> > * Suggested changes 2: Please expand the evaluation or provide your motivation as to why these specific datasets have been selected. Are there no standard benchmark datasets for imputation?
>
> We agree that there was a lack of justification for the choice of the datasets used in the numerical study. We explain now both in the introduction and at the beginning of the numerical study that, to the best of our knowledge, there are no standard benchmark datasets for imputation. We also explain how we chose 7 datasets among those used in Muzellec et al. (2020).
>
> > * Weakness 3: Given that some recent deep learning methods perform poorly, the paper should have a more critical approach and explore/discuss the findings in a more technical way as well.
> > * Suggested changes 3: Can you please investigate the reasons for the poor performance of some recent deep learning methods; see my comment above as well regarding providing a more critical evaluation and discussion of the results.
>
> We agree with the reviewer that the poor performances of recent deep learning methods are surprising. They are not easy to explain but we give two possible reasons in the article. We have added a sentence at the end of section 3 stating that the hyper-parameters of these methods are difficult to choose (we have used the authors' recommendations in our results). We also point out in the conclusion that this poor performance may be due to the nature of the data (tabular numerical data), given that deep learning methods generally perform well on image and natural language processing data.
>
> > Weakness 4: The paper is a bit shallow and includes several sections that are trivial, e.g. explaining autoencoders and denoising autoencoders.
>
> We understand the reviewer's point. This article has been written so that it can be understood and used by people interested in imputing missing data but not necessarily familiar with autoencoders and denoising autoencoders.  For this reason, we believe that explaining these methods is nevertheless useful. Moreover, because the proposed methodology is specific to the imputation of tabular numerical data, we have added in the presentation of the autoencoders and denoising autoencoders (section 2) a few technical details on data normalization and the output layer activation function, specific to numerical tabular data.
>
> We also believe that this theoretical introduction to autoencoders and denoising autoencoders, although trivial,  allows the reader to better understand, theoretically, the benefit of the modified loss function compared to the classic loss function usually applied to pre-imputed data . As already mentioned, the Section 2.1 has been modified to present, hopefully, more clearly the theoretical motivation for the mDAE methodology.
>
> > Suggested changes 4: Please consider additional ablation studies on network architecture choices.
>
> To include this suggestion, we have added in section 3.1 a new figure (Figure 6) with the results of  additional ablation studies for the 6 network architectures, proposed for the choice of the structure (Figure 5).
>
> >  Weakness 5 :  I think that the mDAE method is more of a methodology around how to treat missing values in datasets rather than a new DAE method per se.
>
> We agree with this comment and we now use the word methodology in many places.

---

> > ### Comment · Reviewer_i3J1 · 2024-09-09
> > **Thank you**
> >
> > Thank you for the revised version - I have read the other reviews and responses and I find the current version to have adopted most of the changes suggested. I am happy with that.

---

### Review · Reviewer_S55d · 2024-08-08

**Summary Of Contributions:**

The paper modifies the denoising auto-encoder framework to handle missing values. This is done by simply ignoring missing data points in the reconstruction loss. It carries out a detailed ablation study of its hyperparameters and comparison to existing methods. Here, the method  pesforms sligthly worse than SOTA methods but the modification is shown to improve over a naive DAE approach.

**Audience:**

Yes

**Claims And Evidence:**

Yes

**Requested Changes:**

Abstract
- "The specificities of the proposed [...]". Sentence should be simplified.
- Don't use weasel words: "demonstrate relevance", "good behavior" - It's not really clear what this means without metric or comparison.
- Why not compare on standard metrics, especially in the abstract I'd like to see a metric that I can understand without reading the full paper?

Section 2
- The description in 2.1 is confusing. What is pre-imputed? You say "pre-imputed by column means" but then the first equation shows that we impute with zeros.
- The exposition in equation (3) is not very useful. It's not worth spelling out in as much detail and not worth comparing to in the experiments section.

Section 3
- Are all data types in the test data numerical? Would be good to mention.
- The results of Table 2 could be visualized, e.g similar to Fig. 6
- Do you have an explanation for the modified loss improving results on the climate data? It seems rather suprising.

**Strengths And Weaknesses:**

Strength:
- Relevant problem
- Simple approach
- Reasonable benchmarks

Weakness
- Very limited novelty
- Rather weak empirical results
- The paper spends a lot of time explaining introductory concepts.

---

> ### Author Response · Authors · 2024-08-27
> **Response to reviewer S55d**
>
> We thank the reviewer S55d  for his evaluation of the paper and his insightful comments. Our answers to the reviewers' questions/comments are given below. Note that changes in the revised manuscript are shown in blue.
>
> > Changes requested in the abstract:
> > * "The specificities of the proposed [...]". Sentence should be simplified.
> > * Don't use weasel words: "demonstrate relevance", "good behavior" - It's not really clear what this means without metric or comparison.
> > * Why not compare on standard metrics, especially in the abstract. I'd like to see a metric that I can understand without reading the full paper?
>
> We agree with these remarks. We have modified the abstract by specifying each time the metric used to evaluate or compare the proposed methodology. In particular, we indicate that the results are evaluated with the standard Root Mean Square Error (RMSE) of reconstruction of missing values. We also explain how the new metric  called Mean Distance to the Best (MDB) is computed from the RMSEs of several datasets to globally compare several imputation methods.
>
> > First change requested in Section 2: \
> > The description in 2.1 is confusing. What is pre-imputed? You say "pre-imputed by column means" but then the first equation shows that we impute with zeros.
>
> We agree that the pre-imputation step was not very clearly explained at the beginning of section 2.1. We now specify in section 2. that the mDAE methodology applies to tabular numerical data and that the chosen normalization is therefore standardization (so that the features have a zero mean and unit variance). Pre-imputation by the mean then simply consists of replacing missing values with 0.
>
> > Second change requested in Section 2: \
> > The exposition in equation (3) is not very useful. It's not worth spelling out in as much detail and not worth comparing to in the experiments section.
>
> We understand the reviewer's point about the usefulness of equation (3). We have modified section 2.1 to better explain the difference between the mDAE methodology and the naive approach usually used to impute missing values by applying a DAE to pre-imputed data. This difference can be clearly seen by comparing the loss function (3) used in the naive approach and the loss function (5) that characterizes the mDAE methodology. We therefore believe that equation (3) is useful, especially for people interested in imputing missing data but not necessarily familiar with autoencoders and denoising autoencoders.
> If we understand the reviewer's point about the usefulness of equation (3), we are not sure to understand his point about the usefulness of comparing cost functions (3) and (5) in the experiments. In fact, it seems important to us to compare numerically the results obtained with the modified loss function (5) (and therefore with the mDAE methodology) and the results obtained with the loss function (3) almost always used in previous works (according to the review paper of Pereira et al. (2020)). These results are compared in the second row of Table 2. of the ablation study (section 3.1).
>
> > First change requested in Section 3: \
> > Are all data types in the test data numerical? Would be good to mention.”
>
> We agree that it wasn’t specified clearly enough that the mDAE methodology applies to numerical tabular data.
> We now mention this in several locations of the revised paper.
>
> > Second change requested in Section 3: \
> > The results of Table 2 could be visualized, e.g similar to Fig. 6
>
> We agree that it might be nice to visualize the results of the ablation study in a figure rather than in a table. But it would probably have been necessary to make 4 figures to represent  the 4 last rows of Table 2 and somehow add  the gain in RMSE (results in brackets in Table 2) on these figures. We are not convinced this would have been easier to read, so we have chosen to keep Table 2.
>
> > Third change requested in Section 3: \
> > Do you have an explanation for the modified loss improving results on the climate data? It seems rather surprising.
>
> We agree that it is surprising that results are improved for all datasets except for climate data. We don’t really have an explanation, but this result is consistent with those found by Muzellec et al. (2020) on the same dataset.

---

### Review · Reviewer_gGmc · 2024-08-19

**Summary Of Contributions:**

The paper introduces mDAE for missing data imputation. mDAE is a modified version of DAE (Denoising Auto-Encoder) for overcoming the problem that the latter one has when (pre-)training on fully complete data for later making a robust imputation on the missing values of the dataset. The principal strategy that the authors propose consists of a first phase of zero-imputation on the missing values, which at times, allows to train DAE. Since imputing zero-values on the missing features does not really contribute in any way, the authors skip those values when computing the reconstruction error loss (Eq. 4). Empirical results show the performance of the proposed method within other well-known SOTA techniques also for missing data imputation. Importantly, the performance of mDAE is usually in 3rd position.

**Audience:**

Yes

**Claims And Evidence:**

Yes

**Requested Changes:**

I would like to see critical changes in the paper related to my previous comments that could make me believe that the work provides a technical contribution sufficiently novel to deserve acceptance. But being completely honest with the authors, I see this paper in its current state closer to a ML-conference workshop paper rather than a one over the bar of a journal like TMLR.

**Little question.** I am curious to know why the RMSE is almost always lower in the experiments with 40% than in the ones with 20% missing rate. For instance, Figure 7 vs Figure 8 and Figure 10. This seems kind of counterintuitive, or maybe there is a re-normalization problem with the metrics. Could be that?

**Strengths And Weaknesses:**

**Strengths.**

Despite the work has important flaws in my opinion, particularly on the technical side, I must say that clarity and writing is one of the best parts of the paper. In that regard, it is easy to follow and understand what authors actually propose, so I appreciate that no obscure formulations or derivations hide the contribution of the work, which is understandable for any reader familiar with the missing imputation problem. Another strength of the work is perhaps the spirit of comparing the method with additional SOTA techniques from the last years --- honestly reporting the performance, which in this case is not even the 1st or 2nd with the lowest RMSE on the missing values.

**Weaknesses.**

In my opinion, the contribution and novelty of the work are somehow 'short' for a paper in a journal like TMLR. Being an adaptation of DAE, an already well-known method kind of re-adjusted with zero-value imputation (or mean imputation), which is one of the straightforward methods for missing data since before the 90s is simply not enough for publication in my opinion. I would love to believe that simple things that work really well are greater than complex ones, but in this case --- I don't even see a clear improvement with respect to SOTA, and experiments are certainly not convincing to me. The fact that only 20% and 40% rates are considered is in general a bad sign. It basically quickly brings my mind back to the extremely well-known Figure 1 in the missing data community of the paper from 1993 of Ghahramani and Jordan, where the performance of mean-imputations dramatically falls when the missing rate is approximately higher than 50%. I invite authors to test if this is true also for their method and check what is the actual robustness of it.

Last but not least, the notation and formulation of the paper are somehow misleading, as the notation of latents and outputs is kind-of-inverted wrt the current "canon" used in almost every paper where a latent space is considered.

**References**
Z. Ghahramani and M. I. Jordan (1993) Supervised learning from incomplete data via an EM approach.

---

> ### Author Response · Authors · 2024-08-27
> **Response to reviewer gGmc**
>
> We thank the reviewer gGmc  for his evaluation of the paper and his insightful comments. Our answers to the reviewers' questions/comments are given below. Note that changes in the revised manuscript are shown in blue.
>
> > * Weakness : Being an adaptation of DAE, an already well-known method kind of re-adjusted with zero-value imputation (or mean imputation), which is one of the straightforward methods for missing data since before the 90s is simply not enough for publication in my opinion.
> > * Requested changes: I would like to see critical changes in the paper related to my previous comments that could make me believe that the work provides a technical contribution sufficiently novel to deserve acceptance.
>
> We agree that the mDAE methodology proposed in this article is an adaptation of the classical DAE approach. In the revised version of the article, we have clarified the difference between the proposed mDAE methodology and existing imputation methods based on DAEs, in order to better highlight our contributions.
>
> A first contribution of our work concerns the use of a modified loss function. According to the paper of Pereira et al. (2020), almost all the previous DAE approaches for missing values imputation boil down to calculating the loss function on pre-imputed data. If using a pre-imputed matrix solves the problem of the loss function that is unable to handle missing values, minimizing the reconstruction loss (3) learns the DAE to reconstruct zeros at the locations of the missing values, which is irrelevant. In this paper, we modify the loss function to skip these locations. We numerically validate this finding as shown in the second row of Table 2. We have gathered explanations regarding this contribution in section 2.1.
>
>
> A second contribution of our work is the general and reproducible grid-search methodology for choosing the hyper-parameter and the structure.  Indeed, Pereira et al. (2020) points out that the vast majority of
> previous DAE methodologies used in the field on missing values imputation, do not present justifications for the decisions performed for the choice of the structure and the choice of the hyper-parameters. Furthermore, the results of our ablation study allow us to recommend, when the grid-search approach is too computationally expensive, to choose an overcomplete structure and eventually to choose randomly the value of the hyperparameter mu in [0,1].
>
>
>
> We think that a third contribution of this article is to propose a comparison methodology that is as accurate and reproducible as possible so that other researchers can use it with other datasets or imputation methods using Python codes that will be available on GitHub. We propose a new criterion called MDB (Mean Distance to the Best) which enables an imputation method to be evaluated globally on several datasets. This criterion is now better explained in the paper.
>
>
> While this article may seem straightforward, it has been deliberately written so that it can be understood and used (thanks to the Python code) by people interested in imputing missing data but not necessarily familiar with autoencoders and denoising autoencoders. Note that we have added in the presentation of the autoencoders and denoising autoencoders (section 2) a few technical details on data normalization and the output layer activation function, specific to numerical tabular data.
>
>
> > Weakness: The fact that only 20% and 40% rates are considered is in general a bad sign.
>
> We agree that the choice of 20% and 40% missing data is not clearly justified in the article. There are a large number of possible results when crossing different proportions of missing values with three mechanisms (MCAR, MAR and MNAR). We had chosen to present results for 20% and 40% as the results obtained with the other configurations (percentages and mechanisms of missing values) generally led to the same conclusions. But in response to the reviewer's comment, we have added the results obtained with 10% of MCAR artificial missing values (new Figure 12 in Appendix B).
>
> > Weakness: It basically quickly brings my mind back to the extremely well-known Figure 1 in the missing data community of the paper from 1993 of Ghahramani and Jordan, where the performance of mean-imputations dramatically falls when the missing rate is approximately higher than 50%. I invite authors to test if this is true also for their method and check what is the actual robustness of it.
>
> We agree that the results presented for 20% and 40% missing values do not allow us to verify the robustness of the mDAE method as this proportion increases. We have added Figure 13 at the end of appendix B to visualize the mean RMSE of the mDAE methodology for proportions of missing values ranging from 10% to 90%. This figure shows that the mDAE methodology remains relatively robust up to 70% of missing values.

---

> ### Author Response · Authors · 2024-08-27
> **Second part of response to reviewer gGmc**
>
> > Weakness:  Last but not least, the notation and formulation of the paper are somehow misleading, as the notation of latents and outputs is kind-of-inverted wrt the current "canon" used in almost every paper where a latent space is considered.
>
> We agree that many works use the notation x for the input variable, y for the output variable and z for the latent variable (particularly those based on the EM method). However, in this article we have chosen the notations used by Vincent et al. (2008 ) to introduce the denoising autoencoders and subsequently reused in other works with DAEs (see Pereira et al. (2020)). The classic notations for DAEs are x for the input vector, y for the latent vector and z for the “reconstructed” vector.
>
>
> > Little question: I am curious to know why the RMSE is almost always lower in the experiments with 40% than in the ones with 20% missing rate. For instance, Figure 7 vs Figure 8 and Figure 10. This seems kind of counterintuitive, or maybe there is a re-normalization problem with the metrics. Could be that ?”
>
> Indeed, as the percentage of missing values increases (from 20% to 40% for instance), it becomes more difficult to correctly reconstruct these missing values and the RMSE criterion should increase. But on Figures 7, 8, 9 and 10 (now renumbered 8, 9, 10 and 11), the ordinate does not represent the RMSE criterion but the MDB (Mean Distance to the Best) criterion.  As explained above, this criterion is the mean  over the datasets, of the differences between the RMSE of the considered method and the RMSE of the best method. This MDB criterion does not necessarily increase with the proportion of missing values. But this MDB criterion is useful for ranking the imputation methods according to their ability to reconstruct “on average” all datasets.

---

### Author Response · Authors · 2024-08-27
**General response to reviewers**

We would like to thank the three reviewers for their pertinent comments and suggestions. We are happy that the reviewers found the following strengths to our paper:
* Responds to a relevant problem with a simple approach.
* Is clear and well written.
* Compares the proposed methodology with 8 other imputation methods (including SOTA techniques from the last years) for multiple datasets and missing data scenarios.
* Proposes a "Mean Distance to Best" (MDB) metric for comparing imputation methods across datasets.
* Code will be made available for reproducibility. and has been provided as supplementary material.

In the revised version (modifications are blue), we have taken into account the reviewer’s comments on the weaknesses of the paper. More precisely:

* The abstract has been modified to be more precise, in particular on the metrics used to evaluate and compare methods.
* Introduction has been rewritten to better explain the difference between the proposed mDAE methodology and existing imputation methods based on DAEs, in order to better highlight our contributions.
* Section 2.1 has been modified to present, hopefully, more clearly the theoretical motivation for the mDAE methodology.
* An explanation regarding the choice of the seven datasets used in the numerical study has been added at the beginning of section 3.
* In section 3.1, the comments on the results of the ablation study have been rewritten to better highlight the strengths of the mDAE methodology, and results for other network structures have been added.
* Section 3.2, results for 10% of missing values were added to those obtained with 20% and 40% of missing values.


More detailed responses to comments of the reviewers have been written for each reviewer individually.

---

### Decision · Action_Editor_arbT · 2024-10-25

**Recommendation:** Reject

**Comment:**

The reviewers appreciated the problem of missing data imputation and highlighted as strengths the fact that the paper is easy to read and the proposed modification of the loss is simple. At the same time, they raised important concerns about the lack of rigor in the empirical evaluation, especially when it comes to the limited setting: only certain missing percentage scenarios were considered. Furthermore, the claims that performance are clearly better with mDAE cannot be entirely supported by the empirical evidence, if one looks at metrics such as the RMSE. The analysis under the proposed Mean Distance to the Best does not provide further insights as it compares different RMSE.
On top of that, I highlight how the baselines used and the dataset considered are small-scale and not sufficient for a rigorous empirical analysis.

While the paper presentation improved a lot during the rebuttal, most of the claims remain not fully supported by evidence. And the motivation why one should use mDAE is not clear. As such, the paper is rejected.

**Audience:**

The problem of dealing with missing data is definitely of interest to the TMLR's audience. At the same time, the proposed DAE-based approach is not different enough from existing ways to impute missing data which, in addition to the limited empirical evaluation, makes it less appealing to the community.

**Claims And Evidence:**

The paper introduces a variant of the loss function to train a denoising auto-encoder (DAE) to impute missing values on tabular data. Specifically, the authors propose to not only to apply a DAE to the preimputed data (as previously done in several works), but also to discard entries set to zero. The proposed approach, named mDAE, is compared on seven small-scale UCI datasets against classical simple baselines for data imputation.

The authors claim that the performance of MDAE consistently beats the baselines and that these results generalize to "several" settings including several proportions of missing values, but only two are considered (20 and 40 percent), and to many dataset, but only some small datasets are considered. This leaves open the question whether the claims about performance are supported by enough empirical evidence.

---

> ### Author Response · Authors · 2024-11-04
> **About the final decision**
>
> We thank the action editor for realizing the final decision. However, we are a little disappointed not to have had any feedback from 2 of the 3 reviewers on the revised version of our paper. The only reviewer who responded to us was positive, declaring, "I have read the other reviews and responses, and I find the current version to have adopted most of the changes suggested. I am happy with that.” This unique response, whereas positive, does not seem to have influenced the final decision.
>
> We are also a little surprised by the "claims and evidence" part where it is indicated that "The authors claim that the performance of MDAE consistently beats the baselines". Our conclusion was rather that the numerical study showed good performances for the mDAE methodology but that these performances were not better than those obtained with the standard SofImpute and missForest methods (this was also stated in the introduction). Our method performances were, however, better than those obtained with four recent methods (based on deep learning and optimal transport). We pointed out in the conclusion that these latter might give better results with larger datasets such as image or natural language processing data. The comment "only two are considered (20 and 40 percent)” was also surprising, in the revised version of the paper, we had added more percentages of missing data (Figure 12 in Appendix B). It was in response to reviewer gGmc. However, we probably were not clear enough about that, and we will further improve the clarity of our paper.
>
> We would like to ask if the Action Editor has any recommendations for improving the quality of our work. Would the Action Editor be willing to evaluate a significantly revised version of the manuscript if we successfully incorporate these recommendations?